# Early warning signals in motion inference

Yuval Hart [1,2], Maryam Vaziri-Pashkam [3], L. Mahadevan [1,4,5,6] *

**1** Paulson School of Engineering and Applied Sciences, Harvard University, Cambridge, Massachusetts, United States of America, **2** Department of Psychology, Hebrew University of Jerusalem, Jerusalem, Israel, **3** Section on Neuro-circuitry, National Institute of Mental Health, Bethesda, Maryland, United States of America, **4** Center for Brain Science, Harvard University, Cambridge, Massachusetts, United States of America, **5** Department of Physics, Harvard University, Cambridge, Massachusetts, United States of America, **6** Department of Organismic and Evolutionary Biology, Harvard University, Cambridge, Massachusetts, United States of America

* lmahadev@g.harvard.edu

**Data Availability Statement:** All data files are available at https://github.com/uvhart/EarlyWarningSignalsInMotionInference.

**Funding:** This work was supported in part by the Intramural Research Program of the NIH, National Institute of Mental Health (NCT00001360–Protocol

## Abstract

The ability to infer intention lies at the basis of many social interactions played out via motor actions. We consider a simple paradigm of this ability in humans using data from experiments simulating an antagonistic game between an Attacker and a Blocker. Evidence shows early inference of an Attacker move by as much as 100ms but the nature of the informational cues signaling the impending move remains unknown. We show that the transition to action has the hallmark of a critical transition that is accompanied by early warning signals. These early warning signals occur as much as 130 ms before motion ensues—showing a sharp rise in motion autocorrelation at lag-1 and a sharp rise in the autocorrelation decay time. The early warning signals further correlate strongly with Blocker response times. We analyze the variance of the motion near the point of transition and find that it diverges in a manner consistent with the dynamics of a fold-transition. To test if humans can recognize and act upon these early warning signals, we simulate the dynamics of fold-transition events and ask people to recognize the onset of directional motion: participants react faster to fold-transition dynamics than to its uncorrelated counterpart. Together, our findings suggest that people can recognize the intent and onset of motion by inferring its early warning signals.

## Author summary

Intention inference is one of the fundamental skills that social organisms need to master. Recent studies indicate that people can predict motion onset before it occurs, raising the question of what informational cues underlie this ability. We use data from an antagonistic game between two participants, an Attacker and a Blocker wherein the Blocker is asked to parry the movements of the Attacker. We find that well before the Attacker starts to move, the fluctuations of the Attacker's body about a static pose are accompanied by early warning signals akin to critical transitions in dynamical systems. These early warning signals allow for the anticipation of the Attacker's impending motion before it occurs. Analysis of the early warning signals allows us to characterize the nature of the critical

93-M-0170, annual report MH002035-39) and NSF
STC CCF-1231216 (LM), and a MacArthur
Fellowship (LM). The funders had no role in study
design, data collection and analysis, decision to
publish, or preparation of the manuscript.

**Competing interests:** The authors have declared
that no competing interests exist.

transition, and to simulate it in a simple dot motion experiment. When the dot motion simulations are shown to human participants, we find that they react faster if there is a coming critical transition relative to a transition devoid of the early warning signals. This suggests that people recognize and act upon early warning signals when inferring the onset of motion, and more generally in motor decision making processes.

## Introduction

Timing is everything in many social interactions. This is most clearly seen in motor action, such as fight-or-flight decisions in conflicts, assessing strategies in individual games and sports [1–5], or in team sports [6–8]. Reacting too early leads to mistaken decisions, and reacting too late yields the advantage to the opponent. This naturally confers a benefit to those who have the ability to infer intent of action in the presence of sparse motor signals. Evidence of this is seen in infants who can anticipate the motion of a hand toward a goal [9], and in sports where quick and accurate reactions to extreme motions arise in soccer, basketball, baseball, tennis [1–5, 10–15] etc. These examples collectively suggest that predictions can be made based on the existence of informational cues from body motion in a range of situations involving infants and adults.

Early work on understanding these cues characterized their location [16, 17] and quantified their dynamics [10], in complex real-life sports scenarios like badminton and tennis [2, 11]. However, it might be also beneficial to focus on simpler scenarios where the cues and their dynamics are highly tractable for quantitative analysis. A recent study [18] suggested the existence of preparatory signals for motion onset in a simulated game that has an offensive and defensive move: the Attacker, moved her finger either toward a right or a left target, while the Blocker aimed to reach the same target as fast as possible. The results showed that the Blockers' response times were ∼100 ms quicker compared with the response times to a similar task with a dot on a screen tracking the same motion trajectory as Attackers' finger motion. Furthermore, response times were not dependent on a particular body region or eye gaze [18]. In another experiment, Blockers saw videos of an Attacker and had to indicate the direction of her movement. Interestingly, the response times were ∼100 ms slower when the Blockers saw a version that replaced the frames before motion onset with a static frame compared with an untampered video stream. These results indicate that people are able to infer the onset of finger motion before it actually starts. However, the study did not reveal either the informational cues participants used to infer intention, nor their dynamics.

The onset of intentional motion in the presence of an external cue requires a transition from a stationary state, where the body fluctuates around a stable pose, to a dynamic state, where the body moves swiftly towards a target. This is similar to the behavior of complex systems in the neighborhood of a critical transition associated with slow-fast dynamics, known as a tipping point seen in a range of biological, neuronal, geological, ecological and financial systems [19–26]. In particular, in the context of motor action, Kelso and colleagues showed that human bi-manual coordinated motion demonstrates a critical transition, where anti-phase synchronous motion transitions to in-phase motion as the frequency of motion crosses a threshold frequency [27–30]. A common theme in all these systems is a slow approach of the system to a critical threshold associated with a parameter, after which the system loses stability and abruptly changes its behavior, and evolves exponentially fast. This implies that although the mean behavior of the system before the critical transition changes very slightly, the fluctuations of the system behavior show strong early warning signals of the approach to the stability

threshold [20–23]. This is manifested in the growth of the autocorrelation, an increase in the time of decay of perturbations, and an increase in the variance of the output signal even before motion ensues. Indeed, analyzing the dynamics of the system thus gives clear evidence of the presence of, distance from, and type of critical transition [20, 22, 23, 28]. Previous works have analyzed the characteristics of noise in motor systems and their implications on motor control [31–33]. Here we show that motion fluctuations signal the onset of motion well before it actually occurs. We characterize this signature in terms of a fold-transition event [22–24, 34]. The manifestation of the fluctuations associated with the approach to the critical transition suggests an experimental test—are people capable of cueing in to those signals. By simulating a fold-transition dynamical system as a dot moving on a screen we show that people can act upon the cues of this specific critical transition.

## Materials and methods

All experimental protocols followed the guidelines for the use of human subjects. The Attacker-Blocker experiments (Experiments 1,2) were approved by the Harvard IRB committee (CUHS) and Experiment 3 was approved by the NIH ethics committee. All participants signed an informed consent to participate in the study. **Participants:** 28 (12 females) and 12 (11 female) right-handed participants with normal or corrected-to-normal vision took part in Experiments 1 and 3, respectively. For details on Experiment 2, see Vaziri-Pashkam et al. [18].

### Experimental design

**Experiment 1: Attacker-Blocker dynamics.** In a design similar to Vaziri-Pashkam et al. [18], pairs of participants sat in front of each other separated by a plexiglass screen roughly 63 cm from the participants. Each trial started with the pair placing their finger on a resting spot in front of them. One participant (Attacker) was given instructions via headphones to tap on one of two foam targets pasted on the plexiglass screen. The other participant (Blocker) had to tap on the same target as fast as they could. The Blocker won if she/he reached the target within a certain time window; otherwise, the Attacker was the winner of the trial. The time window was dynamically adjusted for each pair so that the Blocker won in roughly half of the trials. Body movements were tracked with Polhemus Liberty, an electromagnetic position and orientation measuring system with an update rate of 240 Hz. Seven small position-tracking sensors were affixed to the Blockers' right index finger and Attackers' right index finger, right wrist, right elbow, right and left shoulders, waist, and forehead to track the 3D position of these body parts during the interaction. Each pair completed 5-7 blocks of 30 trials. For accurate measures of the response times we used the maximal sampling frequency available. Changing the length of the time windows did not change the current results, see S1 Text.

**Experiment 2: Dot motion dynamics.** For more details of the experimental setup see [18]. Stimuli were back-projected on the Plexiglass screen. Twenty random motion paths were selected from Attacker data collected in a previous experiment [18] and used to create the dot stimulus for this experiment. This way the dot moved on the screen in the same manner as a human Attacker's finger. Each trial started with a dot presented at the bottom of the screen equidistant from the two targets. The dot then moved to one of the two targets. The diameter of the dot started at 0.67 cm and increased to 1.34 cm at the endpoint to mimic the change in the angular size of the finger when it moves towards the screen. The Blocker had to beat the moving dot and tap on the same target as fast as possible. Success was determined in the same manner as in Experiment 1. Subjects completed three blocks of 40 trials.

**Experiment 3: Fold-transition motion detection.** Stimuli were presented on an LCD screen (1920 x 1080, 60 Hz) using an apple MacBook laptop with MATLAB

Psychtoolbox software. The stimulus was a single black circular dot with the diameter of 0.31 degrees of visual angle presented on a dark gray screen. Each trial started with a dot at the center of the screen that moved horizontally to the left or right. Participants were asked to indicate as fast as they could the direction (left/right) of the dot movement. Dot trajectories followed one of 100 simulations of a fold-transition event according to the following dynamical system [23, 34]:

$$\dot{x} \;=\; y + ax^2 - x^3 + \gamma\xi(t) \tag{1}$$

$$\dot{y} \;=\; \epsilon \tag{2}$$

The parameters for the simulations were: $a = 4$, $\gamma = 0.063$. We used a de-trend function such that the dot's motion fluctuates around the origin until the transition occurs. Each motion trajectory was multiplied by a random sign to randomize the dot's direction of motion (either right or left). The time point for the critical transition was randomly set in a range of 350-700 ms from the beginning of the trial. As control, we created 100 motion trajectories with similar noise properties but without the early warning signals of the correlated signal. These motion trajectories of the dot were created by a Heaviside function, rising at a time point between 350-700ms from the beginning of the trial. The time points for transition were matched with the time points of the fold-transition simulations. To these motion trajectories we added random uniform noise with noise amplitude taken from the mean of the 100 fold-transition simulations (see S11 Fig, for examples of the simulated trajectories). Each participant completed 100 trials of the fold-transition simulations and 100 trials of the control trajectories. For each participant we measured the choice of direction and the response times from the beginning of the trial. We chose the values of the $a$ and $\gamma$ parameters, as well as the frame rate and task duration in the simulations to capture the same dynamics and transition time observed in the behavioral experiments of the dyadic Attacker-Blocker interactions. We note that the 2D motion of the dot cannot mimic fully the 3D motion of people but rather serves as a proxy to check whether people are sensitive to the difference between a fold-transition with its correlated noise and an uncorrelated counterpart.

## Analysis

**Early warning signal timing extraction.**   In this analysis, we focused on the velocity curves (v(t)) of the first PC of Attacker's motion as recorded by all sensors excluding the Finger sensor. We used a moving time window (40 ms, 48 ms, and 60 ms—see S3–S5 Figs), to calculate the following quantities: 1) The autocorrelation at lag-1, denoted AR(1) 2) The autocorrelation decay time, denoted $\tau$ (S12 Fig), and 3) The variance for each time window, denoted $\sigma^2$.

For each of these measures, we extracted its transition point according to the following criteria: 1. AR(1): The time point of the sharp rise of the AR(1) curve was set by the first point at which the AR(1) curve crossed a threshold of 0.1. 2. Autocorrelation decay time: The time point of the sharp rise in the autocorrelation decay time was set by the point at which the autocorrelation decay time increased by more than 20%. 3. Variance: The time point of the sharp rise of the variance was set by the first point at which the variance crossed a threshold of $1.5^{*}10^{-4}$.

**Correlation of Attacker's and Blocker's motion-onset with early warning signals.**   For each trial of Attacker-Blocker dynamics, we calculated the relation between the Attacker/Blocker motion onset and the 3 early warning signals. To remove outliers from the data (stemming from false positive detection events that make early warning signals appear sooner than they are), we built a probability density function of the points and took the points that were

inside the 1,6, and 10 density contours. These points were used to calculate the linear regression of the data. From the linear regression results, we extracted the intercept that indicates the average time difference between the Attacker's/Blocker's motion onset and the early warning signals. In the main text we report the results where the threshold density value is set to 6 and contains 81% of the data points to balance outliers exclusion and accurate detection of the early warning signals. In the S1 Text we report the results for the more inclusive case where the threshold density value is set to 1 and the region of interest contains 96% of the data points, and the more restrictive case where the threshold density value is set to 10 and the region of interest contains 60% of the data points. Across all the different choices of thresholds, the early warning signals of the autocorrelation sharp rise and autocorrelation decay-time rise occur between 103ms-149ms before Attackers' motion begins (see S1 Text, and S13 Fig).

**Variance fitting.** To calculate the power-law divergence of the variance (see Eq 5) for each Attacker-Blocker dynamics trial we took the 100 ms before and after the divergence of the variance event. To reduce the noise in the variance curves we averaged the variance across each block's 30 trials (synchronized by the divergence point), yielding a total of 85 variance divergence curves. We partitioned each curve by a time window of size 48 ms and an offset of 4 ms and fitted each variance curve with a non-linear model $log(\sigma^2) = -log(b) + n\,log(t^* - t)$, where b, n, and $t^*$ are fitting parameters of the model. We then chose the fitting parameters that provided the best fit across all time windows. For the results of the fitting curves for all of the variance curves, see S9 Fig, and for the results for other time window sizes see S10 Fig.

## Results

Our experimental setup follows the same experimental procedure presented in [18]. In each trial, Attackers were instructed to move their right finger to either a right target or a left target (by a cue to their headset). Simultaneously, Blockers had to guess the Attackers' target by their motion and reach it as fast as they could (Fig 1a). The Attackers had seven sensors attached to their body at the following locations: head, torso, left shoulder, right shoulder, right elbow, right wrist and right finger (Fig 1b). The Blockers had one sensor at their right finger. A total of 14 pairs participated, each pair had between 5-7 blocks, and each block constituted 30 trials

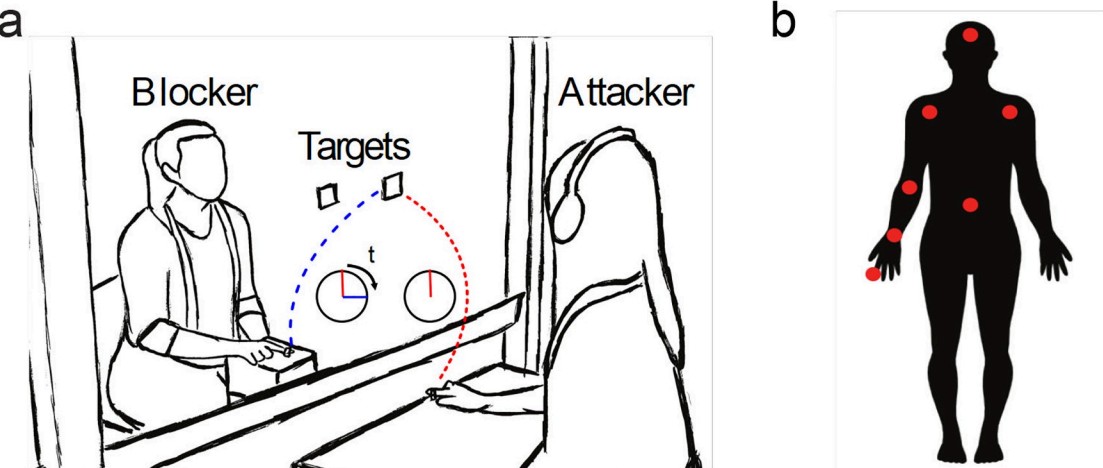

**Fig 1. Experimental setup. a)** A schematic of an Attacker-Blocker trial. The Attacker starts a move towards either the right or left target at some instant of time, and the Blocker's goal is to reach the same target as the Attacker as fast as possible, i.e. to minimize the delay *t*. **b)** Schematic of location of the seven sensors on the Attacker's body.

(a total of 85*30 = 2550 trials across all pairs). Using high-speed video (recording at 240 Hz, see Methods) we analyzed the 3d motion of the Attacker and Blocker at high spatial resolution (spatial resolution 1 mm, see Methods).

## Attacker body motion signal is dominated by center of mass motion

We first characterized the information in the Attacker's body motion using principal component analysis (PCA) on the six body sensors (excluding the finger sensor). In accord with previous findings which showed that no specific body part carries the preparatory signals [18, 35], we found that all body sensors contribute equally to the first PC of motion (mean loadings of each sensor's coordinate and 95% CIs: Wrist = 6.7%, 95% CI = [5.8,8.1], Elbow = 6.3%, 95% CI = [4.2,7.8], Right Shoulder = 6.3%, 95% CI = [4.1,7.6], Left Shoulder = 4.5%, 95% CI = [0.4,6.9], Torso = 4.9%, 95% CI = [0.7,6.8], Head = 4.7%, 95% CI = [0.5,6.8]). The equal loadings of the first PC suggest that the first PC acts as a center of mass motion (see S1 Fig).

## Attacker body autocorrelation rises before motion ensues

Since the Attacker transitions from a resting state to a moving one, we ask which signals serve as early cues for the Attacker's decision to move. Since the fluctuations of the body might serve as cues about intent, we measure the autocorrelation and variance of movement velocity as a function of time, using a running window of size 40 ms (changing window size shows similar results, see S1 Text). For each time window, we calculated the following measures of the Attacker's body velocity ($v(t)$):

1. The autocorrelation at lag-1 (denoted AR(1)):
   $AR(1) = \Sigma_{t_i}(v(t_i) - v_0)(v(t_{i-1}) - v_0)/\Sigma_{t_i}(v(t_i) - v_0)^2$, which measures the extent to which the signal resembles itself after a shift of one data point.

2. The autocorrelation decay time (denoted $\tau$): $\tau = arg_n R(n) = R(0)/e$, where
   $R(n) = \Sigma_{t_i}(v(t_i) - v_0)(v(t_{i-n}) - v_0)/\Sigma_{t_i}(v(t_i) - v_0)^2$, and $arg_n$ chooses the time scale $n$ that holds the equation $R(n) = R(0)/e$, and thus measures the time scale for the autocorrelation to decay to $1/e$ of its maximal value.

3. The variance across the time window (denoted $\sigma^2$): $\sigma^2 = \frac{1}{n-1}\Sigma_{t_i}(v(t_i) - v_0)^2$ (see Methods).

We found that the AR(1) signal shows a sharp rise well before motion onset. AR(1) rise is marked by the transition from negative to positive values. A linear regression between the rise time of AR(1) and the onset time of Attackers' finger motion across all trials showed that on average AR(1) sharp rise precedes finger motion by 136 ms ($\Delta t$: mean ± ste = 136 ± 2 ms, see also S3 and S4 Figs).

Similarly, the autocorrelation decay time rises sharply well before motion onset. We defined the rise in the autocorrelation decay time as the first time point where the decay time increases by 20% between consecutive time windows. The 20% fold change threshold was set to filter out noisy detection events while maintaining accurate detection of a rise in the autocorrelation decay time (see S1 Text). A linear regression between the rise time of the autocorrelation decay time and the onset time of Attackers' finger motion across all trials showed that on average the sharp rise in the autocorrelation decay time precedes finger motion by about 130 ms ($\Delta t$: mean ± ste = 131 ± 2 ms, Fig 2a, and see S3 and S5 Figs).

Contrary to the autocorrelation signals, the variance increases only slightly before finger motion onset. We defined the time of the variance rise as the time point where variance crosses a threshold defined by the maximal variance levels within the first 100 ms of motion across

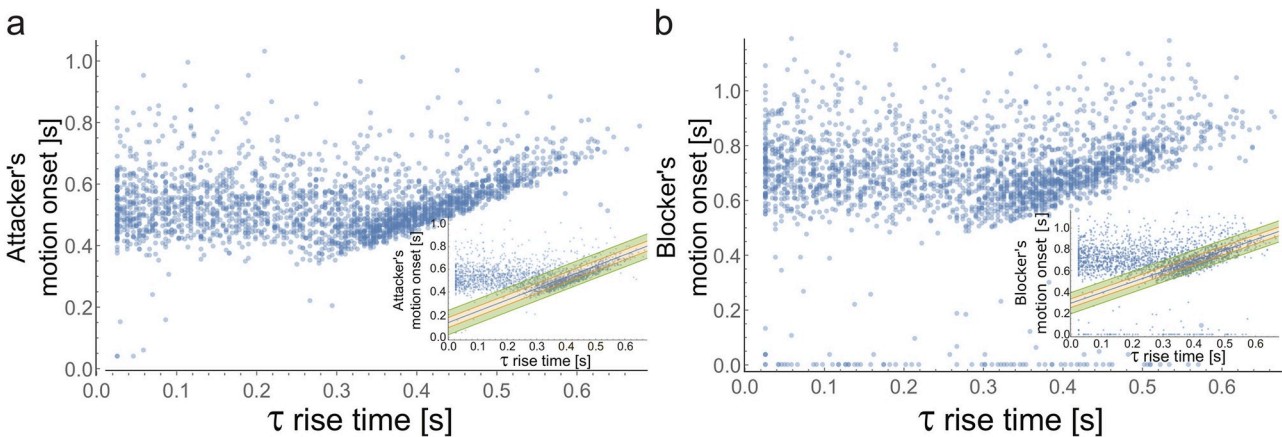

**Fig 2. The increase in the decay time of the autocorrelation signal predicts Attacker's and Blocker's motion onset. a)** Attacker's finger motion onset timing as a function of the timing of the increase in the autocorrelation decay time. **Inset**, For the linear regression calculations, we focused on the highest density regions of the data points (covering 81% of the data points, see S1 Text for other thresholds). Shown are the regression line (solid blue), 50% CI (solid oragne) and 90% CI (solid green) lines. $\Delta t$: mean ± ste = 131 ± 2 ms. **b)** Blocker's finger motion onset timing as a function of the timing of the increase in the autocorrelation decay time. **Inset**, For the linear regression calculations, we removed false positive events of the autocorrelation decay time rising earlier than 200 ms from the beginning of the trial and focused on the top highest density regions of the data points (covering 64% of the data points, see S1 Text). Shown are the regression line (solid blue), 50% CI (solid oragne) and 90% CI (solid green) lines. $\Delta t$: mean ± ste = 293 ± 2 ms.

participants' trials. A linear regression between variance rise time and the onset time of Attackers' finger motion across all trials showed that on average variance rise does not precede finger motion (see S3 and S6 Figs).

Could these early warning signals be related to the early motion of the body before finger motion ensues? To test this we compared the body's center of mass motion to finger motion. For each trial, we determined the onset of motion as the time when the speed of motion first exceeded $0.05 cm/sec$ (following the same definitions as in [18], see also S1 Text). We found that the center of mass of body motion started 50 ms before finger motion started (mean difference: 52ms, 95% CI = [46, 57], $p < 10^{-5}$, effect size = -0.33, see S1 and S2 Figs). Therefore, body motion onset, which happens around 50ms before finger motion, cannot account as the driver of the early warning signals, occurring about 130ms before finger motion.

Our results show that there is a sharp rise in AR(1) and a sharp rise in the autocorrelation decay time, both of which provide early warning signals of the Attacker's decision to start moving. Importantly, these signals are derived from the fluctuations of the body in its static pose, even before motion of body and finger ensues. However, it may still be the case that while these signals precede Attacker's motion, the Blockers are unaware of them and react in response to other signals. To test this, we next determine the relation between the Attackers' early warning signals and the Blockers' reaction times.

## Early warning signals are correlated with Blocker motion

We used the Blocker's time of response as defined in Pashkam-Vaziri et al. [18], i.e. it is the first time point where Blocker's finger motion crosses 0.05 $cm/s$ in the same direction as the Attacker's finger motion. A linear regression between the AR(1) sharp rise timing and the Blockers' onset time of finger motion across all trials indicated a high correlation with an average time difference of 309ms ($\Delta t$: mean ± ste = 309 ± 2 ms, $p < 10^{-5}$, S3 and S4 Figs). Similarly,

a linear regression between the timing of autocorrelation decay time sharp increase and the Blockers' onset time of finger motion across all trials indicated a high correlation with an average time difference of 293ms ($\Delta t$: mean ± ste = 293 ± 2 ms, $p < 10^{-5}$, Fig 2b, and S3 and S5 Figs). Thus, autocorrelation rise time and the increase in autocorrelation decay time are highly correlated with Blockers' response times. Furthermore, the time difference between the two explains well the quick response of the Blockers, anticipating the Attacker's motion onset. We note that a linear regression between variance rise and the Blockers' onset time of finger motion shows a much shorter time difference of 65$ms$ and thus is less probable to act as a signal for the Blockers' response ($\Delta t$ mean ± ste = 65 ± 6 ms, $p < 10^{-5}$, S3 and S6 Figs).

In an additional experiment (Experiment 2), Pashkam-Vaziri and colleagues [18] showed participants a moving dot on a screen. The dot dynamics were taken from actual Attacker's finger motion. We analyzed the dot motion to reveal no early warning signals in both the autocorrelation and the decay time of the autocorrelation (see Methods, and S3 and S7 Figs). Consistent with this, the participants' response times were slower compared to participants who saw a video of the entire body movement (Reaction times difference of ∼100-120 ms [18]). To test whether the lack of early warning signals might be a consequence of the sampling of dot motion from Attackers' finger motion, we further analyzed the Attackers' finger motion in Experiment 1. We found that finger motion of the Attacker did not show early warning signals; Its AR(1) signal started off positive and thus the decay time of the autocorrelation starts at high values from the beginning of the trial and does not rise further. Analysis of Attackers' finger motion indicates no significant correlation between Attackers' motion onset and the rise in AR(1) or in the autocorrelation decay rate of finger motion (AR(1) signal: linear regression slope ± ste = 0.06±0.06, p = 0.35, Autocorrelation decay time: linear regression slope ± ste = -0.04±0.03, p = 0.23). Thus, lack of the early warning signals was accompanied by slower response times of the Blockers.

The early warning signals in the autocorrelation of the Attacker's body fluctuations indicate the existence of a critical transition and raises the question of its nature, which we now turn to.

## Body fluctuations before motion-onset are characteristic of a fold-transition (saddle-node bifurcation)

To quantify how the Attacker switches from a stationary state to a dynamic one, we consider a minimal model that can describe the transition as a slow-fast dynamical transition. Given change of the system's state from stable to dynamic one, the simplest structurally stable critical transition of this type is a co-dimension one bifurcation known as a *fold-transition* or a saddle-node bifurcation [22, 24, 34]. Defining the accumulation of "decision momentum" over time by a variable $y(t)$ which changes on a slow timescale ($\epsilon$), and the velocity of the Attacker ($v(t)$) by the variable $x(t)$ which changes on a fast timescale, we write the dynamic equations for the saddle-node bifurcation as

$$\dot{x} \quad = \quad y + x^2 + \xi(t) \tag{3}$$

$$\dot{y} \quad = \quad \epsilon \tag{4}$$

where $\xi(t)$ is a zero-mean Gaussian noise term that models the uncorrelated fast dynamical noise in the system. When $y < 0$, Eq 3 has a pair of static solutions, of which the stable one represents the static state of the Attacker. As $y$ increases, the system approaches a critical transition—When $y$ is positive, Eq 3 has no steady-state solution, and only dynamic motion is possible. Thus, a fold-transition system can describe the Attacker's switch from a stable, static pose to a dynamical one. To test whether the Attacker's transition from static to dynamic

motion follows a fold transition model, we consider the behavior of the variance of the fast variable, $x(t)$, $\sigma(t) = \langle (x(t)-\langle x(t)\rangle)^2\rangle^{1/2}$, where $\langle a(T)\rangle = \frac{1}{T}\int_0^T a(t)dt$, close to the transition point. Near the critical transition time point, $t^*$, theory suggests that for a fold transition, the variance should diverge according to the scaling law [22, 23, 34]

$$\sigma^2 \propto (t^* - t)^{-1/2} \qquad (5)$$

The inverse square root relation between the variance and the time interval to transition reflects the system's fluctuations as it reaches the departure point from the stable manifold on which the scaling relations are: $x \sim \sqrt{y}$ (Eq (3)) and $y \sim t$ (Eq (4)) [22, 23].

For each Attacker, we calculated the time point at which the variance crosses the threshold ($T^* = arg_t\ \sigma^2(t) > 1.5^*10^{-4}$, see S1 Text) and extracted variance curves 100ms before and 100ms after that time point. We then averaged the variance trajectories in each block of trials (30 trials in each block), resulting in a total of 85 variance trajectories. Next, we fitted each of the variance curves in the neighborhood of the transition to a general power-law divergence model of the type $log(\sigma^2) = -log(b) + n\ log(t^*-t)$, where b, n, and $t^*$ are fitting parameters of the model (Fig 3, and S9 Fig). We find that the scaling exponent of variance divergence is close to -0.5

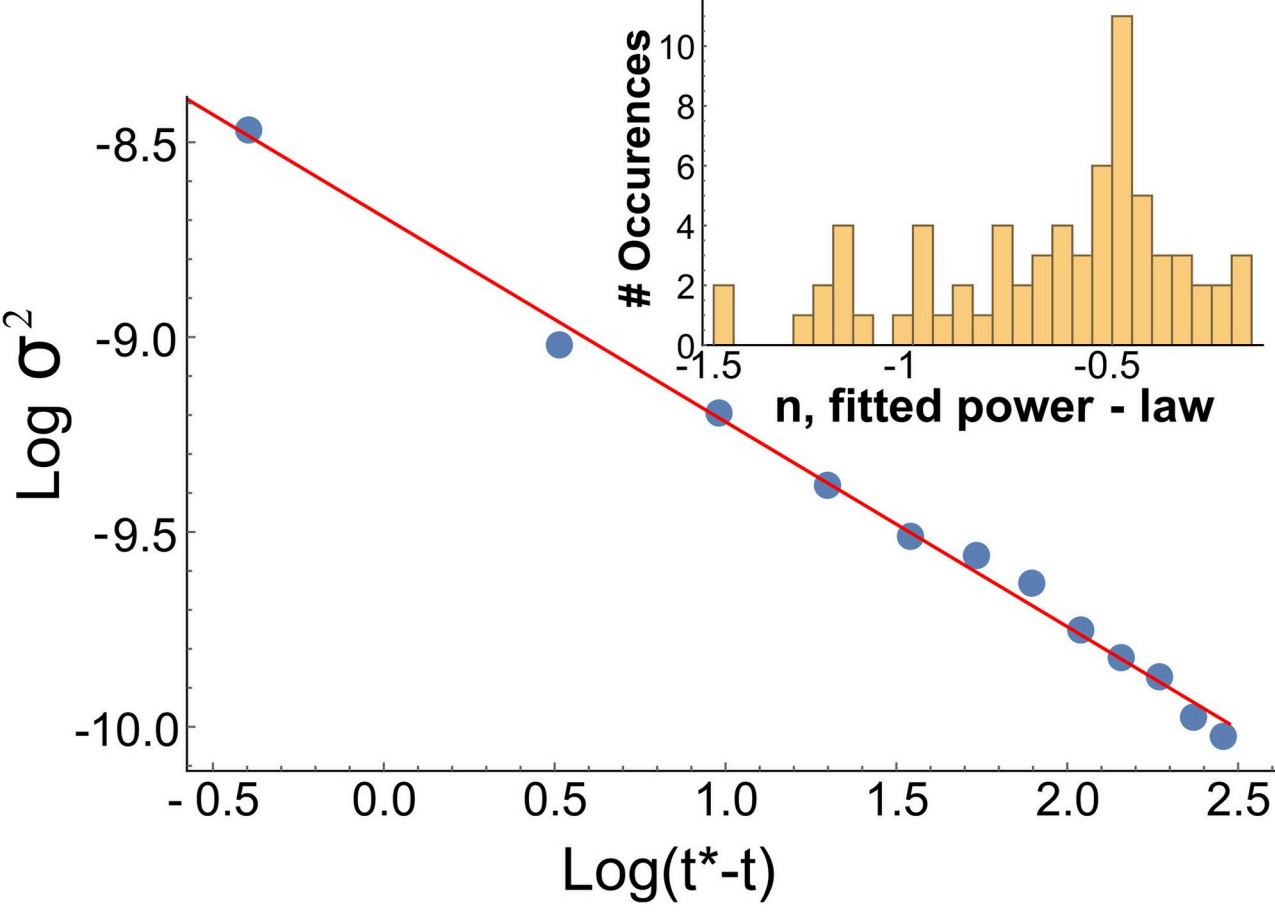

**Fig 3. Variance of the movement velocity diverges as one approaches the critical point following fold-transition characteristics.** An example of fitting model of $log(\sigma^2) = -log(b) + n\ log(t^* - t)$ to a variance trajectory. Shown is the fit over a window of 12 consecutive time points (Red solid line). The fit parameter for the power-law (n) in this example is -0.53. **Inset**, Histogram of all fitted power-law parameters smaller than 1.5 (for the entire histogram, see S1 Text). Median fitted power-law is: n = -0.54, 95% CI = [-0.66, -0.5].

(median: n = -0.54, 95% CI = [-0.66, -0.5], Fig 3, and S10 Fig). It is worth emphasizing that the scaling exponent for the variance is a quantitative measure for the type of critical transition, e.g. other co-dimension one critical transitions such as the pitchfork, and Hopf transitions are associated with the variance diverging according to: $\sigma^2 \propto (t^* - t)^{-1}$ [22, 23]. Thus a measure of the nature of the critical transition can be gleaned by determining the scaling exponent of the variance, and suggests that a fold transition model describes the Attacker's switching dynamics reasonably well.

## Simulated fold transition data for motion-onset produces faster reaction times from participants

Given our ability to extract the nature of the critical transition from observations, i.e. a fold-transition, we reverse the question and ask if people can identify the early warning signals that are induced by a simulated fold-transition model with noise. In Experiment 3, we showed participants (N = 12) a moving dot on a screen that followed one of two simulated scenarios: 1) the dynamics associated with a critical fold-transition event and 2) an uncorrelated analog.

To derive the simulated data of the fold-transition we solved the following dynamical system 100 times to create 100 trajectories of 1d motion (see S11 Fig and S1 Video):

$$\dot{x} \quad = \quad y + ax^2 - x^3 + \gamma\xi(t) \tag{6}$$

$$\dot{y} \quad = \quad \epsilon \tag{7}$$

The quadratic term $a$ sets the steady-state position the system reaches after the transition [34], while the cubic term causes the output to saturate after the transition, and the $\gamma\xi(t)$ term represents the inherent noise in the system (see Methods). Readouts from the simulation were spaced 10 ms apart for a total duration of a second and a half, while the location of the critical transition was randomly varied between trials. As expected, preceding the critical transition from the static state to the dynamical state, the resulting trajectories are characterized by a sharp rise in the autocorrelation and an increase in autocorrelation decay time.

As a control, we also created a fast transition from a stationary state to dynamic motion in a random dynamical system, but mimicking the same critical time (denoted $t^*$) as in the fold-transition simulations, and added uniform random noise ($\xi(t)$) with noise amplitudes that matched the noise levels of the fold-transition simulations (see S11 Fig and S2 Video):

$$x(t) = a\Theta(t - t^*) + \xi(t) \tag{8}$$

where $\Theta(x)$ is the Heaviside function. The resulting uncorrelated motion trajectories showed similar noise amplitudes but in the absence of non-trivial dynamics before the transition, there were no early warning signals of the AR(1) and the autocorrelation decay time showed no interesting signal.

Participants were shown blocks of trials from these two stimuli in random order, wherein they saw a dot on the screen. They were asked to respond to the onset of directional motion of the dot. Participants estimated the direction of movement accurately for both stimuli types, and were not significantly different between the two conditions (fold-transition accuracy rate = 99%, uncorrelated transition accuracy rate = 99%, Mann-Whitney: $U(143) = 55$, $p = 0.24$). In Fig 4a, we see that the responses to the fold-transition simulation were significantly faster than the responses to the uncorrelated stimuli ($\Delta t$ mean ± ste = 100 ± 27 ms, Mann-Whitney: $U(143) = 20$, $p = 0.002$, effect-size = 0.72, see S1 Table). We further find that the aggregated response times distributions of the fold-transition and the uncorrelated transition are significantly different (Kolmogorov-Smirnov test, $p < 10^{-5}$, see Fig 4b) and that the

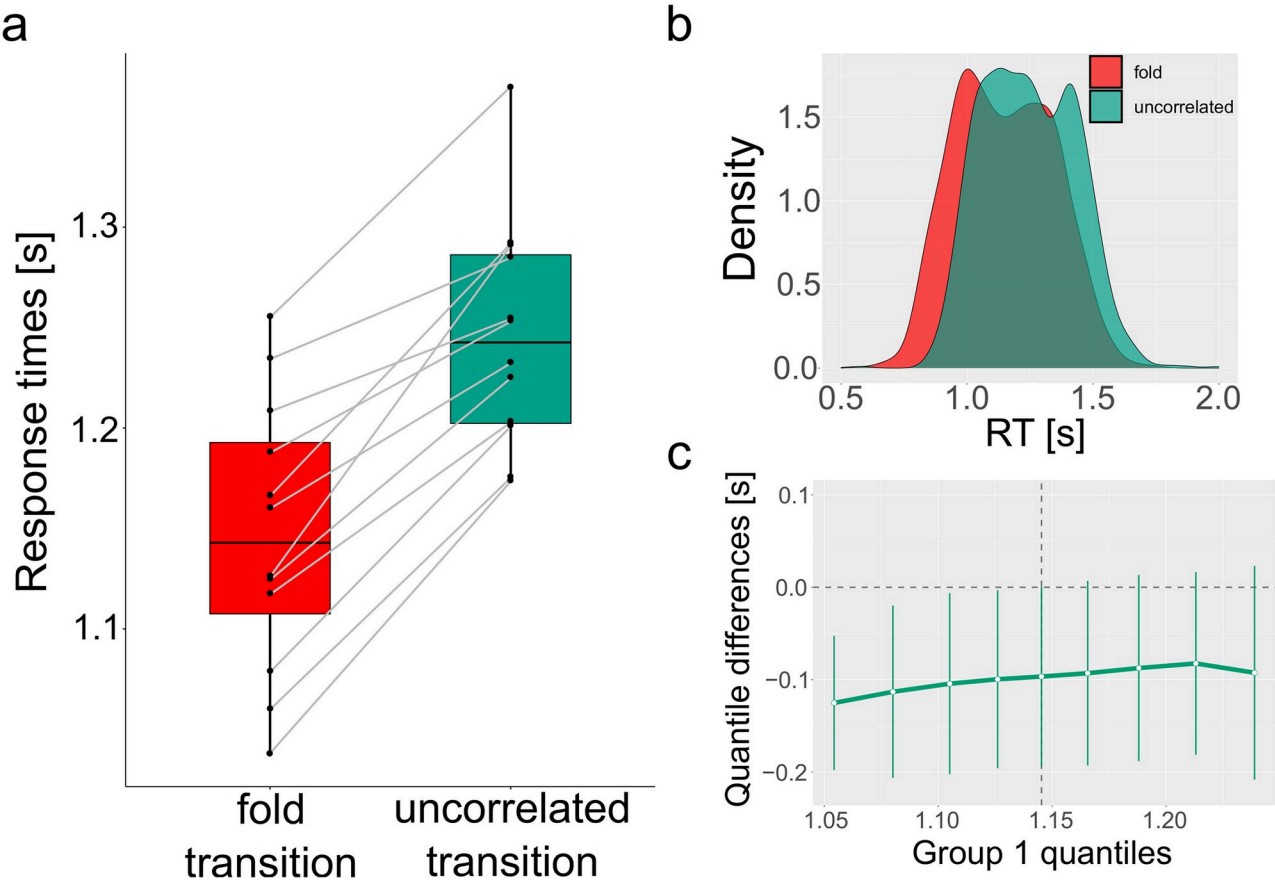

**Fig 4. Participants act upon the early warning signals in fold-transition events. a)** Participants' response times for motion of a fold-transition event (Eqn (6-7)) are significantly faster than their response to an uncorrelated version (Eq (8)) of the transition ($\Delta t = 100ms \pm 27ms$, Mann-Whitney: $U(143) = 20$, $p = 0.002$, effect-size = 0.72). **b)** Participants' response times distribution for the fold-transition (red) and uncorrelated transition (green). **c)** The shift function of decile differences between the fold-transition and uncorrelated transition response times distributions.

shift-function of decile differences [36, 37] shows a constant difference around 100ms between the two distributions (Fig 4c). We therefore conclude that participants could act upon the early warning signals induced by a fold-transition event.

## Discussion

Since much of behavior is predicated on movement, inference of motion onset in response to signals from the outside provides a cognitive advantage to organisms. A strategy that favors an ability to sense early warning signals would thus be evolutionary advantageous and suggests looking for informational cues in motor action. In two experiments, we have shown that in a two-person interaction, the Blocker perceives early warning signals based on the fluctuations of the Attacker's body, well before the Attacker begins to move.

Our study further quantifies this by showing that Attackers' switch from static pose to dynamic motion is preceded by two cues which are the hallmarks of early warning signals for critical transitions—A rise in the lag-1 autocorrelation and a rise in the decay time of the auto-correlation. These two cues are highly correlated with the motion onset of the Blockers. By extracting the power-law associated with the divergence of the variance near the critical point,

we find that Attackers' decision to move is consistent with a fold critical transition. We then show that participants in a third experiment react faster (while maintaining similar accuracy) to motion-onset driven by a fold-transition preceded by correlated noise which shows changes in the autocorrelation signals, compared with motion-onset preceded by uncorrelated noise, which does not show any changes in the autocorrelation signals.

The reaction by Blockers to early warning signals raises the question of the underlying mechanism that governs this early detection ability. Our data analysis process indicated that the detection of the increase in the autocorrelation decay time was less affected by noisy measurements than the AR(1) signal (See Fig 3, and compare S3, S4 and S5 Figs). Computationally, it might suggest a preference for the measurement of the decay time of the autocorrelation as a cue for Blockers' action. We note that the observed 300ms time difference between the early warning signals and Blockers' response time might relate to the P300 signal in EEG literature [38, 39]. Further studies of our experimental paradigm for social interactions in EEG setting (similar to Refs [40, 41]) could indicate if that is indeed the case.

Although we have focused on the early warning signals associated with the Attacker's motor decision making process, it is likely that a similar critical transition in motion-onset is produced by the Blocker as well. We hope that future studies might consider the two-body problem that couples two dynamical systems with fold-transition-like normal forms that model the Blocker's decision to move contingent on the Attacker's putative movement in a game theoretic setting [15].

Our work has explored the informational cues that are inferred by the Blocker from the Attacker's motion in a relatively simple setting, complementing a cohort of previous works on the dynamical aspects of joint-action and social motor coordination [30, 42–44] all of which involve aspects of motor decision making and intention inference in a range of settings. For example, participants react differently to grasping movements depending on the social cues they infer from their partner's motion [45, 46] and team sports players move according to the affordances incurred by their teammates and their opponents [47, 48]. Building on our work, we suspect that further study of possible co-representations in Attacker and Blocker's motoric and neuronal activity can indicate whether motor simulation [7, 49, 50] plays a role in these interactions, thus allowing us to elucidate the biological mechanisms that may underlie accurate inference of intention [51–55]. Our study highlights the need to focus on the detection and response to early warning signals in the fluctuations of the signal as they are indicators of critical transitions underlying intention and decision making. Mapping the conditions under which humans and other animals can or cannot recognize and act upon different types of critical transitions would shed new light on the environments they were evolved in and the computational cognitive mechanisms they have developed.

## Supporting information

**S1 Fig. Attacker first PC of body motion is its center of mass which precedes finger motion.**
(TIF)

**S2 Fig. Motion of the Attacker's center of mass precedes motion of the Attacker's finger.**
(TIF)

**S3 Fig. Attacker's and Blocker's motion onset correlate with the early warning signals.**
(TIF)

**S4 Fig. Autocorrelation at lag-1 sharp increase correlates with Attacker's and Blocker's finger motion onset also at time windows of 48 ms and 60 ms.**
(TIF)

**S5 Fig. Autocorrelation decay time sharp increase correlates with Attacker's and Blocker's finger motion onset also at time windows of 48 ms and 60 ms.**
(TIF)

**S6 Fig. Variance sharp increase correlates with Attacker's and Blocker's finger motion onset also at time windows of 48 ms and 60 ms.**
(TIF)

**S7 Fig. Autocorrelation decay time threshold balances accuarate detection and false positives.**
(TIF)

**S8 Fig. Finger dot motion does not show early warning signals.**
(TIF)

**S9 Fig. Variance sharp rise near the transition point shows a fold-transition type of divergence.**
(TIF)

**S10 Fig. Scaling power-law for different window sizes shows a divergence with power law of -0.5.**
(TIF)

**S11 Fig. Trajectory samples for the fold-transition simulations and their uncorrelated counterpart.**
(TIF)

**S12 Fig. As the transition point approaches, the autocorrelation decay time rises sharply.**
(TIF)

**S13 Fig. The different density thresholds taken to screen outliers in the Attackers' relation with the early warning signals.**
(TIF)

**S1 Text. Supporting information.**
(PDF)

**S1 Video. Video samples for the simulation of motion-onset following a noisy fold-transition with correlated noise preceding it.**
(MP4)

**S2 Video. Video samples for the simulation of motion-onset with uncorrelated noise preceding it.**
(MP4)

**S1 Table. Response times (mean±ste) for fold-transition and uncorrelated motion simulations per participant.**
(XLSX)

## Author Contributions

**Conceptualization:** Yuval Hart, L. Mahadevan.

**Data curation:** Yuval Hart, Maryam Vaziri-Pashkam.

**Formal analysis:** Yuval Hart, Maryam Vaziri-Pashkam.

**Investigation:** Yuval Hart, Maryam Vaziri-Pashkam, L. Mahadevan.

**Project administration:** L. Mahadevan.

**Supervision:** L. Mahadevan.

**Visualization:** Yuval Hart.

**Writing – original draft:** Yuval Hart, Maryam Vaziri-Pashkam, L. Mahadevan.

**Writing – review & editing:** Yuval Hart, Maryam Vaziri-Pashkam, L. Mahadevan.

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
