## [Decision Letter · Decision Letter 0]

14 Sep 2019

Dear Dr Mahadevan,

Thank you very much for submitting your manuscript, 'Early warning signals in motion inference', to PLOS Computational Biology. As with all papers submitted to the journal, yours was fully evaluated by the PLOS Computational Biology editorial team, and in this case, by independent peer reviewers.

Well done. The paper made a very good impressions on the reviewers and the editors, and we are confident that this will also be the case for the readership of the journal.

There are some aspects of the manuscript that should be improved, and that the reviewers nicely caught.

We would therefore like to ask you to modify the manuscript according to the review recommendations before we can consider your manuscript for acceptance. Your revisions should address the specific points made by each reviewer and we encourage you to respond to particular issues Please note while forming your response, if your article is accepted, you may have the opportunity to make the peer review history publicly available. The record will include editor decision letters (with reviews) and your responses to reviewer comments. If eligible, we will contact you to opt in or out.raised.

- Supporting Information uploaded as separate files, titled 'Dataset', 'Figure', 'Table', 'Text', 'Protocol', 'Audio', or 'Video'.

We hope to receive your revised manuscript within the next 30 days. If you anticipate any delay in its return, we ask that you let us know the expected resubmission date by email at ploscompbiol@plos.org.

Sincerely,

Daniele Marinazzo

Deputy Editor

PLOS Computational Biology

Daniele Marinazzo

Deputy Editor

PLOS Computational Biology

[LINK]

Reviewer's Responses to Questions

**Comments to the Authors:**

Reviewer #1: The manuscript details three experiments and a set of simulations exploring the subtle movement dynamics that human actors are sensitive to when anticipating or predicting another actor’s movement intention. Using an actor-defender stimulus-repose paradigm, the results of empirical experimentation and simulation work revealed that defending actors are sensitive to early warning signals in an actor’s center of mass (or principle component motion time-series) that occur approx. 150 ms before the onset of an actor’s task specific movement. Moreover, the dynamics of an attacker’s motion near the point of movement transition corresponds to the dynamics of a fold-bifurcation, a fundamental property of nonlinear dynamical systems.

This is a fantastic piece of work. The experimental paradigms and stimuli employed, although simply, and well-conceived and provide a valid method of determining the subtle movement dynamics that help observers (defenders) anticipate or infer the action intention of an observed opponent. The data analysis and modelling techniques are highly novel with respect to the phenomena investigated, with the overall findings likely to have a transformative impact with the fields of human-movement science, perception-action and joint-action and social motor control and coordination. The writing is excellent, with the method, figures and supplementary material well-constructed and easy to follow.

My only concern relates to the discussion section, which is largely just a summary of the findings, and does not effectively situate the work within the wider join-action and social (multiagent) motor coordination and action observation literature. Research on action observation and cooperative and competitive performance has a long history in psychology, cognitive science and human-movement science, with a particular increase in focus over the last two decades. A core debate within this literature, is whether the a purely neuro-cognitive or representational approach (e.g., Prinz, Sebanz and Knobloch and colleagues co-representational/motor resonance account of joint action and action observation; as well as the work on mirror neurons and motor simulation by numerous neuroscientists and cognitive neuroscientists) is required to explain such phenomena in comparison to a more ecological-dynamical (informational) account (e.g., Araujo Duarte and colleagues, Christina Becchio and colleagues, Schmidt and Richardson and colleagues; Yuji Yamamoto and colleagues, in addition to Kelso and colleagues). While these two general approaches to action observation and coordination are perhaps best understood as complementary, the current work is highly relevant to these theoretical issues and the work might have more general impact if the authors better situate this work within the context of this wider literature. Note, that I am not saying that the authors need to make any strong theoretical claims, or even articulate any theoretical conclusions one way or the other, but rather better detail how the work relates to the previous work on the dynamics of action observation and prediction that exists within the join-action and social motor control literature.

I appreciate that the authors would probably like a list of key references, but this literature is rather large and I do not wish to bias the authors assessment of this literature or what they feel should or should not be include. The authors should also see the above listed names as a rather limited set of relevant researchers and there are many more that could be considered.

Reviewer #2: # Review

## Summary of the article

The article addresses an interesting paradox brought by recent studies in motor behaviour: people are able to predict others' motion trajectories even before the start of their movement. The authors used a simple and elegant paradigm called the "antagonistic game" where one player —the Blocker— has to parry on the movement of another player –the Attacker. In a first experiment, they demonstrated how the fluctuations of the centre of mass of the Attacker's body are actually giving early signs of his/her ensuing movement 150ms before it starts. The authors then demonstrate how those fluctuations show characteristics of a saddle-node bifurcation. And finally, simulating similar dynamics and presenting them to human participants, they show how the simulated movements also induce faster reaction times.

## Main impressions of the article

The study is very interesting and, despite its technical aspects, remains concise and clearly written. I think it has the sufficient impact for PLoS Computational Biology ensuing few minor corrections.

## Comments and suggestions

Note: it would have been nice to use a one column presentation, and, to add page and line numbers to ease the review.

Title, Abstract, and Introduction are well written and the former covers well existing literature.

In the autocorrelation formula, describes what is "arg_n".

To what extend the different measure of autocorrelation and variance across time window is modulated by sampling frequency?

The authors define "the rise of autocorrelation decay time" with a decrease of "20% between two consecutive time windows" but do not explicit why.

The 347ms time difference observed between the AR(1) sharp rise and Blockers' start time remind the P300 in the EEG literature. That may be interesting to discuss.

Fig. 2: Add the linear regression line and confidence interval. Maybe the two panels can be combined using two colours, thus helping comparison between the two measures.

This is not clear why the AR(1) signal of Attacker's finger starting positive implicates that decay time only makes a sharp rise at the beginning of each trial.

Would it be possible to briefly explain how (3) is derived from (1-2)?

Fig. 4: Getting a sense of distributions is important. Violin plot or even the recent raincloud plots would be perfect here.

Adding videos of the two types of simulated dynamics would be great to get a subjective sense of the difference.

This is not clear why the increase of autocorrelation decay time is less noisy than the AR(1)

This is not clear why AR(1) would necessarily need to store in memory two timeseries (cannot be implemented through dynamical systems without a memory per see). Is there a formal proof and a reference supporting this claim?

In the method, this is not clear how the dot projected on a 2D screen can mimic human dynamics which is in 3D.

How the authors have chosen the parameters of the dynamical systems (alpha & gamma)? Does the choice of those values strongly impact the results?

**Have all data underlying the figures and results presented in the manuscript been provided?**

Reviewer #1: Yes

Reviewer #2: No: No mention of public repository for the data and code.

PLOS authors have the option to publish the peer review history of their article (what does this mean?). If published, this will include your full peer review and any attached files.

Reviewer #1: No

Reviewer #2: Yes: Guillaume Dumas

---

## [Editor Report · Decision Letter 1]

2 Mar 2020

Dear Dr. Mahadevan,

Thank you very much for submitting this revised version of your manuscript "Early warning signals in motion inference" for consideration at PLOS Computational Biology.  Thanks for addressing the issues raised. We are likely to accept this manuscript for publication, after you have addressed an outstanding point related to the reaction time reporting and stats

 1. please report all data points, not just the mean and error barshttps://onlinelibrary.wiley.com/doi/full/10.1111/ejn.13400   and consider the use graphical methods such as https://onlinelibrary.wiley.com/doi/abs/10.1111/ejn.13610   2. caution has to be taken in applying t-tests and the like to skewed distributions such as RTs, see https://www.annualreviews.org/doi/pdf/10.1146/annurev.publhealth.28.082206.094100 

https://www.biorxiv.org/content/10.1101/383935v3

We apoligize to bringing these issues to your attention only at this stage, as a result of increased awareness of possible issues in reporting of the results.

On the other hand we are confident that this will improve the reception and presentation of a very nice study.

thanksPlease prepare and submit your revised manuscript within 30 days. If you anticipate any delay, please let us know the expected resubmission date by replying to this email. 

Sincerely,

Daniele Marinazzo

Deputy Editor

PLOS Computational Biology

[LINK]
---

## [Editor Report · Decision Letter 2]

24 Mar 2020

Dear Dr Mahadevan,

We are pleased to inform you that your manuscript 'Early warning signals in motion inference' has been provisionally accepted for publication in PLOS Computational Biology.

Best regards,

Daniele Marinazzo

Deputy Editor

PLOS Computational Biology

Daniele Marinazzo

Deputy Editor

PLOS Computational Biology

---

## [Editor Report · Acceptance letter]

28 Apr 2020

PCOMPBIOL-D-19-01376R2 

Early warning signals in motion inference

Dear Dr Mahadevan,

I am pleased to inform you that your manuscript has been formally accepted for publication in PLOS Computational Biology. Your manuscript is now with our production department and you will be notified of the publication date in due course.

With kind regards,

Laura Mallard
